# Modeling the Impact of Exogenous Boosting and Universal Varicella Vaccination on the Clinical and Economic Burden of Varicella and Herpes Zoster in a Dynamic Population for England and Wales

**DOI:** 10.3390/vaccines10091416

**Published:** 2022-08-28

**Authors:** Oluwaseun Sharomi, Ilaria Xausa, Robert Nachbar, Matthew Pillsbury, Ian Matthews, Tanaz Petigara, Elamin Elbasha, Manjiri Pawaskar

**Affiliations:** 1Merck & Co., Inc., 126 E Lincoln Ave, Rahway, NJ 07065, USA; 2Wolfram Research Inc., 100 Trade Center Drive, Champaign, IL 61820, USA; 3Independent Researcher, Rahway, NJ 07065, USA; 4Merck Sharpe and Dome (UK) Ltd., 120 Moorgate, London EC2M 6UR, UK

**Keywords:** varicella, herpes zoster, vaccination, cost-effectiveness, dynamic modeling

## Abstract

Universal varicella vaccination (UVV) in England and Wales has been hindered by its potential impact on exogenous boosting and increase in herpes zoster (HZ) incidence. We projected the impact of ten UVV strategies in England and Wales on the incidence of varicella and HZ and evaluated their cost-effectiveness over 50 years. The Maternal-Susceptible-Exposed-Infected-Recovered-Vaccinated transmission model was extended in a dynamically changing, age-structured population. Our model estimated that one- or two-dose UVV strategies significantly reduced varicella incidence (70–92%), hospitalizations (70–90%), and mortality (16–41%) over 50 years. A small rise in HZ cases was projected with UVV, peaking 22 years after introduction at 5.3–7.1% above pre-UVV rates. Subsequently, HZ incidence steadily decreased, falling 12.2–14.1% below pre-UVV rates after 50 years. At a willingness-to-pay threshold of 20,000 GBP/QALY, each UVV strategy was cost-effective versus no UVV. Frontier analysis showed that one-dose UVV with MMRV-MSD administered at 18 months is the only cost-effective strategy compared to other strategies. HZ incidence varied under alternative exogenous boosting assumptions, but most UVV strategies remained cost-effective. HZ vaccination decreased HZ incidence with minimal impact on the cost-effectiveness. Introducing a UVV program would significantly reduce the clinical burden of varicella and be cost-effective versus no UVV after accounting for the impact on HZ incidence.

## 1. Introduction

Varicella, or chickenpox, is an acute and highly contagious disease caused by the varicella-zoster virus (VZV). While usually mild and self-limiting, varicella can lead to severe complications, hospitalizations, and death in rare cases [1,2]. The most common complications are bacterial infections, decreased platelets, arthritis, hepatitis, pneumonia, and encephalitis. They can occur in all age groups but are more likely to occur in neonates, immunocompromised persons, and adults [3]. Globally, varicella is one of the most common childhood infectious diseases, with incidence estimates ranging from 2 to 16 cases per 1000 persons [1,4,5]. It poses a significant economic and caregiver burden due to its high incidence and potential complications [6]. VZV has a second clinical manifestation, herpes zoster (HZ), a painful vesicular rash occurring later in adulthood upon reactivation of the latent form of the virus in nerve ganglions [1].

Live-attenuated VZV vaccines are active immunizing agents that offer protection against VZV infection. Randomized clinical trials (RCTs) and observational studies have established the safety and efficacy of these vaccines for the prevention of varicella [7,8,9] and have been licensed and added to routine childhood vaccination schedules in many countries worldwide [10]. Universal varicella vaccination (UVV) programs have led to significant reductions in disease incidence [11,12]. For example, in the United States (US), where it was introduced in 1995, varicella incidence declined by 98% from 1990 to 2016 [11]. In addition, varicella hospitalizations and deaths declined by 99% from 2012 to 2016 compared to the pre-vaccination period from 1990 to 1994 [11].

The World Health Organization (WHO) recommends UVV in countries that can maintain vaccination coverage rates of ≥80% [13]. One dose is recommended to reduce mortality and severe morbidity from varicella but is insufficient to limit virus circulation and prevent outbreaks [13]. A two-dose schedule has higher effectiveness and is therefore recommended in countries where the programmatic goal is to further reduce the number of cases and outbreaks.

In the United Kingdom (UK), varicella vaccination is not part of the routine childhood immunization program and is only offered to those in close contact with individuals at high risk of varicella and its complications [14]. Consequences of this policy impact different age groups. General practitioner (GP) consultation rates for varicella are highest among children between 1 and 3 years of age (39.7 per 1000 person-years in 2014) [15]. Most hospital admissions for varicella were in children under 10 years of age (79.4% during 2004–2013) [16]. In contrast, most HZ admissions (71.5%) over the same timeframe occurred in adults 60 years of age or older [16]. 

In 2010, the Joint Committee on Vaccination and Immunization (JCVI) in England recommended against a UVV program in children or a combined childhood varicella and adult HZ vaccination program due to its potential impact on exogenous boosting and HZ incidence [17]. It has been hypothesized that UVV, by reducing circulating wild-type VZV, also reduces exogenous boosting, the phenomenon by which re-exposure to VZV boosts protective cell-mediated immunity and delays reactivation of latent VZV. A decrease in exogenous boosting could potentially cause an increase in the incidence of HZ or shift the incidence of natural varicella to older age groups who are more likely than younger individuals to experience severe illness, leading to hospitalization and higher costs [17]. Prior modeling studies that evaluated the impact of UVV on HZ incidence with different assumptions for exogenous boosting found that UVV implementation may not be cost-effective when evaluated over a short time horizon in the UK [16,18].

Since the JCVI’s recommendation, several real-world studies have advanced our understanding of the impact of UVV and exogenous boosting. A recently published self-controlled case series study of 9604 UK adults with both household exposure to varicella and an episode of HZ showed that exposure to varicella is associated with a 33% reduction in risk of HZ over 20 years, suggesting that the impact of exogenous boosting may be lower than predicted in previous models [19]. Another database study examining 20 years of experience with UVV in the US found that the transient increase in the incidence of HZ predicted by the models was not observed [20]. The annual incidence of HZ in adults increased at approximately the same rate in the years before and after implementing the UVV program. The increase in HZ incidence prior to UVV implementation may be explained by historical demographic changes that were not considered by previous models [21,22].

Models that consider the interplay between exogenous boosting and demographic changes may more accurately predict the impact of varicella vaccination programs on HZ incidence. The objective of this study was to estimate the long-term clinical and economic impact of UVV strategies in England and Wales as well as their cost-effectiveness, considering exogenous boosting scenarios based on recent real-world evidence, in a population with a dynamically changing age structure.

## 2. Materials and Methods

### 2.1. Description of the Dynamic Transmission Model

A deterministic, age-structured, continuous-time, nonlinear dynamic transmission model in the form of differential equations was developed (details provided in Appendix A). The model uses the Maternal-Susceptible-Exposed-Infected-Recovered-Vaccinated structure [23], extended to include health states that represent the reactivation of latent VZV causing HZ outbreaks. The age-defined compartments were defined to capture the demographic, epidemiological, behavioral, and economic inputs as well as vaccination schedules of interest. The health states tracking vaccinated persons were further subdivided to differentiate between those who received one or two varicella doses. The model incorporated a dynamically changing population using over 50 years of historical data on time-dependent mortality, fertility, and migration rates. Thus, the impact of changing demography on disease incidence can be measured, and the model reflects the range of available historical data on the epidemiology of VZV in England and Wales, especially varicella seroprevalence.

### 2.2. Demographic and Epidemiological Parameters

Demographic and epidemiological parameters are provided in Table 1. The model was calibrated to age-stratified varicella seroprevalence [24,25,26], and HZ incidence [18] in England and Wales. Varicella- and HZ-related death were fitted against mortality data summarized in Brisson and Edmunds [24] through calibration as well (details in Appendix A).

Most parameters related to varicella and HZ reactivation were taken from a modeling study by Schuette and Hethcote [23]. The model stratifies varicella-infected individuals into latent and infectious health states. Susceptible individuals may become infected at a rate governed by contact rates between age groups [31,40]. The same average latent period was used for natural and breakthrough infection [23]. However, the likelihood of a vaccinated individual infecting others following breakthrough varicella was reduced due to a shorter infectious period and lower relative infectivity of breakthrough varicella than natural varicella. The relative infectivity of breakthrough varicella was based upon a 1997–2001 population-based active surveillance study in which vaccinated cases were half as contagious as unvaccinated cases [36]. 

Persons previously infected with VZV can undergo HZ reactivation as immunity against HZ wanes. Individuals who were successfully vaccinated against varicella can undergo HZ reactivation as well, which results in recovery with lifelong immunity or death. The model assumes that individuals can benefit from exogenous boosting, providing temporary partial immunity throughout their lifetime with parameters derived from recent real-world evidence on the impact of contact with persons with infectious varicella on rates of HZ in the UK [19]. Using a predictive model (see Appendix A), the waning period following natural or breakthrough varicella and the proportion of contacts leading to exogenous boosting were calculated [19].

### 2.3. Vaccine Properties

Vaccine properties are described in the Appendix A. Vaccine failure rates (defined as individuals who did not seroconvert within 42 days of vaccine administration) were drawn directly from RCTs with 10 years of follow up (4% for Varivax^®^ [Merck Sharp & Dohme LLC, Rahway, NJ, USA, V-MSD] and 5% for Varilrix^®^ [GlaxoSmithKline, Wavre, Belgium, V-GSK]) [8,41,42]. The first and second dose take is the rate at which a complete immunological response is induced following the first dose and the second dose if the first dose does not provide complete immunity, respectively [43]. We estimated take and duration of protection using deterministic compartmental models to simulate clinical trials of one- or two-dose varicella vaccination with V-MSD and V-GSK. Our model estimated that 90.3% (95% confidence interval (CI): 87.8–92.9%) of the cohort gained permanent protection from breakthrough varicella after the first dose of V-MSD compared to 61.7% (95% CI: 58.2–65.3%) after the first dose of V-GSK. We further estimated that a total of 97.0% (95% CI: 95.2–98.8%) and 93.8% (95% CI: 92.2–95.4%) of the cohort were permanently protected after two doses of V-MSD and V-GSK, respectively [43].

### 2.4. Vaccination Strategies

Ten UVV strategies are included in the model (detailed in Appendix A). Since JCVI is considering an additional visit for measles, mumps, rubella (MMR) vaccination at 18 months [44], we included varicella vaccination strategies that aligned with the MMR schedule at 12 and 18 months. By aligning with MMR, the total number of pediatric office visits are not increased, and it also provides the opportunity to use combination MMRV vaccine in a single vaccine. In the single-dose strategies, children received a quadrivalent formulation at 18 months (ProQuad^®^ [Merck Sharp & Dohme LLC, Rahway, NJ, USA, MMRV-MSD] or Priorix-Tetra^®^ [GlaxoSmithKline, Wavre, Belgium, MMRV-GSK]). In the 2-dose strategies, children received a monovalent formulation at 12 months (V-MSD or V-GSK), followed by a second dose of the monovalent or the quadrivalent formulation at 18 months or 4 years of age, and with or without catch-up vaccination. The first and second catch-up dose for teenagers who missed childhood varicella vaccination was aligned with the human papillomavirus vaccine schedule. Catch-up vaccination was only offered during the first two years of the program. Vaccination coverage rates were set to 91% for the first dose and a second dose given at 18 months, 88% for a second dose at 40 months [45], and 87% for catch-up vaccination based on the tetanus, diphtheria, pertussis, polio vaccine (Tdap/IPV) [46].

### 2.5. Health Utilities and Cost

Age-specific utility weights for healthy individuals, individuals with natural and breakthrough varicella, as well as with HZ, are provided in Appendix A.

The model includes vaccine-related costs, direct medical care costs, and when taking the societal perspective, the costs of work lost to varicella and HZ. All costs were adjusted for inflation and expressed in 2020 GBP. The estimated cost per dose for MMRV-MSD (GBP 46.63) and MMRV-GSK (GBP 55.13) are calculated from the average list prices in Germany [47], Spain [48], and Switzerland [49], and converted to British pounds due to lack of list pricing for MMRV in the UK. Price per dose for V-MSD (GBP 30.28) and V-GSK (GBP 27.31) were obtained from the NICE British National Formulary for Children [50]. A cost for vaccine administration (GBP 9.80) was also included [51]. Direct cost components are provided in Appendix A. For varicella and HZ, the costs of outpatient care included consultations with general practitioners (GP) and the cost of treatments (for cases seen in GP offices). It was assumed that every HZ outbreak resulted in treatment of some type, and as such, HZ-associated outpatient costs were applied to all cases. For varicella and HZ inpatient care, the cost of treatments was added to the cost per admission and applied to hospitalized cases. Health care utilization and associated costs were assumed to be the same for natural and breakthrough varicella and wild-type and vaccine-type HZ. Parameters for workdays lost due to VZV are provided in Appendix A. 

### 2.6. Model Outcomes and Cost-Effectiveness Analysis

The key model outcomes projected over time included varicella cases, HZ cases, and related outpatient visits, hospitalizations, and deaths. Outcomes were categorized as those related to natural and breakthrough varicella, and those related to HZ.

Incremental cost-effectiveness ratios (ICERs) were computed by comparing each UVV strategy to no UVV from both the payer and societal perspectives. In addition, a frontier analysis was conducted to inform decisions as to which strategy or strategies should be considered when choosing among all feasible ones. In this analysis, strategies dominated by one or several others are removed. The ICER of non-dominated strategies on the frontier is performed by determining whether the additional QALYs gained by the next strategy along the frontier is worth the incremental cost. The threshold for cost-effectiveness of GBP 20,000 per QALYs in the UK was used [52]. For all analyses, costs and QALYs were discounted at 3.5% annually [53,54]. The impact of UVV was assessed over a 50-year time horizon in the base case.

### 2.7. Sensitivity Analysis

To examine the impact of uncertainty in key vaccine and cost parameters on the cost-effectiveness of the UVV strategies that are on the frontier, parameters (or sets of related parameters) were varied one at a time in a deterministic sensitivity analysis (DSA). In addition, a probabilistic sensitivity analysis (PSA) using 500 variates was conducted to assess the variability in ICERs relative to the no-UVV scenario as a function of the uncertainty in key parameters relating to the calibration of HZ incidence and varicella seroprevalence, vaccine strategy, and costs. 

Scenario analyses were also conducted to investigate the robustness of the cost-effectiveness results under alternative assumptions. Since the impact of UVV on HZ incidence is dependent on the assumptions used for exogenous boosting, three alternative scenarios were evaluated based upon the most recent literature and previous assumptions used in modeling studies. In scenario EB1, the waning period of HZ immunity was 24.4 years, with the effectiveness of contacts being age-dependent (75%, 71%, 57%, and 32% for those 0–59, 60–69, 70–79, and >80 years of age, respectively) [55]. In scenario EB2, the waning period of HZ immunity was 81.3 years, and 100% of contacts were effective. In scenario EB3, exogenous boosting was eliminated; the waning period of HZ immunity was 81.3 years, and no contacts were effective. In a fourth scenario, we assessed the impact of including HZ vaccination, which was introduced in the UK national immunization program (NIP) in 2013 for adults 70–79 years of age [56]. Finally, we conducted a sensitivity analysis using a 1.5% annual discount rate for long-term effects, consistent with NICE recommendations for vaccines [57].

## 3. Results

### 3.1. Calibration

Detailed results of the model calibration are provided in Appendix A. The model reproduced the dynamically changing population age structure compared to historical data from selected years between 1971 and 2018 (Appendix A). With the calibrated age structure, the model aligned well with historical VZV seroprevalence data from 1978, 1992, 2004, and 2007 (Appendix A). For HZ incidence, the historical data from 1986 through 2006 were not stratified by year, so a year-to-year comparison was not possible. However, the assessment of the model fit using assumed demographics for 1995 showed that model-based projections are bounded by the historical HZ data and followed the general pattern of increasing incidence with age (Appendix A).

### 3.2. Health Outcomes

#### 3.2.1. Varicella

All proposed UVV strategies are projected to rapidly reduce total varicella incidence after introduction. Two years after introducing a UVV program, incidence decreased from approximately 1100 cases per 100,000 person-years without UVV to around 135–210 cases per 100,000 person-years depending on UVV strategy (Figure 1a). Breakthrough varicella incidence rose after vaccine introduction and stabilized at values ranging between 10 and 247 cases per 100,000 person-years. Breakthrough cases were lower for two-dose strategies compared to one-dose strategies. The highest number of breakthrough cases was projected for Strategy B (one dose MMRV-GSK) (Figure 1b).

The percentage reduction in varicella health outcomes (total varicella cases and varicella-related outpatient visits, hospitalizations, and deaths) after 50 years with each UVV strategy in comparison to no UVV is shown in Figure 2. Total varicella cases are estimated to decline by 70.0–92.1%, while the total number of varicella-related outpatient visits and hospitalizations is estimated to decline by 71.7–92.1% and by 70.5–89.8%, respectively, over a 50-year period. Varicella-related deaths are estimated to decline by 15.8–40.9% post-UVV introduction. For all health outcomes, percentage reductions were higher with two-dose strategies compared to single-dose strategies. Catch-up vaccination among adolescents 13 and 14 years of age offered during the first two years of UVV provided modest additional reductions in varicella cases. Strategies C and D, which are two-dose short-interval strategies with catch-up vaccination, resulted in the highest-percentage reductions in varicella cases (92.1% and 90.2%, respectively). The strategy with the highest-percentage reduction in varicella cases was Strategy C (two-dose short-interval strategy with V-MSD, including catch-up vaccination). There were no differences in health outcomes between Strategies G and I for MSD vaccines and between Strategies H and J for GSK vaccines, since the efficacy of monovalent varicella vaccines and the efficacy of the varicella component of quadrivalent vaccines were considered equivalent.

#### 3.2.2. Herpes Zoster

The projected incidence of HZ between 2022 and 2072 following UVV in the base-case scenario is depicted in Figure 3a. Without UVV, HZ incidence peaked in 40 years (2062), rising from 382 to 408 cases per 100,000 person-years (a 6.8% increase). Following UVV, HZ incidence peaked during the first 20–22 years (2042–2044) at around 402–409 cases per 100,000 person-years, a 5.3–7.1% increase compared to pre-UVV rates. HZ incidence began to decline under all UVV strategies around 2045 and fell below the pre-UVV incidence rate by 2050. By 2072, HZ incidence was estimated at around 328–335 cases per 100,000 person-years, a 12.2–14.1% decrease compared to the pre-UVV period.

### 3.3. Cost-Effectiveness

#### 3.3.1. Base-Case Scenario

Results from the cost-effectiveness analysis comparing the UVV strategies over 50 years in England and Wales from a payer perspective and societal perspective are provided in Table 2. Implementation of the UVV program resulted in total direct costs ranging from GBP 2.12B to GBP 2.74B, compared to GBP 1.72B without vaccination. 

Each UVV strategy resulted in higher costs from the payer perspective, but also resulted in QALYs gained (fewer QALYs lost) due to the decrease in total varicella infections and HZ cases. Compared to no UVV, QALYs gained range from 54,854 (Strategy B: single-dose with MMRV-GSK) to 61,826 (Strategy C: a two-dose strategy including a catch-up with V-MSD). The ICER for each UVV strategy compared with no UVV ranged between 6809 GBP/QALY gained and 16,698 GBP/QALY gained. Thus, each strategy was cost-effective compared to no UVV when using the willingness-to-pay threshold of GBP 20,000 per QALY gained [52]. Strategies on the frontier included Strategy A (MMRV-MSD 18 months), Strategy G (V-MSD 12 months; MMRV-MSD 40 months, with catch-up), and Strategy C (V-MSD 12 and 18 months, with catch-up). All other strategies were dominated by one or more of these three strategies. If a threshold of 20,000 GBP/QALY was applied to the frontier strategies, only Strategy A would be deemed cost-effective.

From a societal perspective, Strategy A is cost-saving compared to no UVV, as QALYs were gained while reducing total costs by GBP 86M. All other strategies remained cost-effective, with each ICER at or below GBP 8003 per QALY gained vs no UVV. From the societal perspective, the same UVV strategies are on the frontier (Strategies A, G, and C). However, unlike the payer perspective, no UVV was strongly dominated by Strategy A, and thus does not lie on the frontier. Using the threshold of 20,000 GBP/QALY gained, only Strategy A would be deemed cost-effective, as the ICERs for Strategy G versus Strategy A and for Strategy C versus Strategy G exceed the threshold.

#### 3.3.2. Deterministic Sensitivity Analysis

Figure 4 shows the robustness of the ICERs for the strategies on the frontier from the payer perspective to one-way deterministic changes in parameters on costs and vaccine properties. In each plot, the resulting upper and lower bound values of the ICER are shown, with the most influential parameters displayed at the top. For Strategy A versus no UVV (Figure 4a), the cost of the vaccine and treatment for varicella infection and HZ reactivation are the most influential parameters. The vaccine coverage, take, and waning rate also influenced the ICER, but to a much lesser degree. In each case, the ICER remained well below 10,000 GBP/QALY gained versus no UVV. For Strategy G versus Strategy A (Figure 4b) and Strategy C versus Strategy G (Figure 4c), vaccine coverage, take, and cost are influential. However, the ICERs for Strategy G and Strategy C remained above 50,000 GBP/QALY gained and 1,000,000 GBP/QALY gained, respectively.

#### 3.3.3. Probabilistic Sensitivity Analysis

The PSA was conducted on the base case to evaluate the uncertainty in the ICER resulting from variation in the calibration and uncertainty in parameters. Results of the PSA showed that the base-case results are robust. Strategies A, G, and C were compared to no UVV from a payer perspective. For Strategy A and Strategy G, all variates led to ICERs below 20,000 GBP/QALY gained. For Strategy C, 97.4% of the variates led to ICERs below 20,000 GBP/QALY gained. The spread of the points from the randomly generated parameter sets surrounding the base-case estimate of expected costs and QALYs and the likelihood of being cost-effective across a range of willingness-to-pay thresholds for the three vaccine strategies are shown in Appendix A, respectively.

#### 3.3.4. Scenario Analyses

Results of the scenario analysis are presented in Table 3. Three scenarios were explored where boosting varies by duration of HZ immunity after varicella infection and by the percentage of varicella contacts that boost against HZ. In scenarios EB1 (waning period of HZ immunity is 24.4 years; age-dependent effectiveness of contacts) and EB2 (waning period of HZ immunity is 81.3 years; 100% effective contacts), peak HZ incidence for the UVV scenarios occurred around the same period (from 2042–2044 and from 2043–2046, respectively) compared to the base case (from 2042–2044). However, HZ incidence for the UVV scenarios remained higher than HZ incidence for no UVV for a longer time for EB1 (where UVV becomes lower between 2054–2059) and EB2 (between 2060–2067) compared to the base case (between 2041–2050) (Figure 3b,c). Thus, in both scenarios (and in the base case), UVV ultimately led to fewer HZ cases due to the diminishing pool of individuals at risk of HZ. The reduction in HZ cases following UVV was immediate in scenario EB3 (waning period of HZ immunity is 81.3 years, 0% effective contacts), when exogenous boosting was eliminated (Figure 3d). In scenario EB1, the incremental costs increased while the QALYs gained decreased, leading to higher ICERs compared to the base case. From the payer perspective, the ICERs ranged from 8864 GBP/QALY with Strategy A to 20,410 GBP/QALY with Strategy D. In scenario EB2, where exogenous boosting was assumed to be even higher, ICERs were higher and ranged from 12,041 GBP/QALY with Strategy A to 26,290 GBP/QALY with Strategy D. When the role of exogenous boosting was removed (EB3), ICERs decreased for all UVV strategies and ranged from 4331 GBP/QALY with Strategy A to 12,192 GBP/QALY with Strategy D. From the societal perspective, all ICERs were below GBP 20,000 for all UVV strategies and exogenous boosting scenarios. While Strategy A no longer dominated a no-UVV strategy under the EB1 and EB2 scenarios, the ICERs were only 1605 GBP/QALY and 6075 GBP/QALY, respectively. In scenario EB3, 6 of the 10 vaccination strategies dominated no UVV, with lower costs and more QALYs gained.

We also assessed the impact of HZ vaccination, which was introduced in the UK NIP in 2013 for adults aged 70–79 years old [56]. Total costs were higher for all UVV strategies (including no UVV) as the cost of the HZ vaccination program was included. Compared to the base case without HZ vaccination, the number of varicella cases and outcomes were slightly lower (<1% reduction) for each UVV strategy. The introduction of an HZ vaccine had a slightly greater impact on reducing the incidence and burden of HZ (Figure 3e). ICERs improved slightly as HZ vaccination resulted in lower incremental costs (3% to 8% reduction) and greater incremental QALYs gained (1% to 2% higher) for all UVV strategies compared to no UVV. For example, ICERs of 6183 GBP/QALY and 6,809 GBP/QALY were obtained for Strategy A compared to no UVV in the presence and absence of an HZ vaccination program, respectively.

Finally, the results of the sensitivity analysis with a 1.5% discount rate for health outcomes are provided in Table 3. Total costs for each strategy remained the same, but incremental QALYs increased. Consequently, the ICERs for each strategy were lower than in the base case and ranged from GBP 4408 to GBP 10,778 per QALY from the payer perspective and from cost saving to GBP 5166 per QALY from the societal perspective.

## 4. Discussions

We adapted a dynamic transmission model to estimate the long-term health impact and cost-effectiveness of implementing a UVV program in England and Wales, considering different scenarios of exogenous boosting in a population with a dynamically changing age structure. Overall, ten vaccination strategies were considered using monovalent and quadrivalent formulations of MSD or GSK vaccines with one or two doses, short or medium intervals, and with or without catch-up vaccination. 

All strategies substantially reduced the clinical and economic burden of VZV. Over 50 years, both one-dose strategies with quadrivalent MMRV formulations administered at 18 months of age with assumed coverage of 91% were projected to reduce varicella mortality by 16–32% and varicella incidence by 70–87% (with similar reductions in outpatient visits and hospitalizations). All two-dose strategies, where the first dose was administered at 12 months and the second dose at 18 or 40 months, were projected to have higher reductions in varicella mortality (37% to 41%) and incidence (89% to 92%). Breakthrough varicella cases were higher for one-dose strategies compared to two doses, and were highest for MMRV-GSK; this could be attributed to lower efficacy of one-dose MMRV-GSK [7]. Our findings are consistent with the literature as the efficacy of two doses is higher than that of one dose [7,8]. Two-dose strategies are typically recommended to reduce breakthrough cases and prevent outbreaks [13]. 

While the benefits of UVV on reducing the burden of varicella are well-established, the potential impact of UVV on exogenous boosting and HZ incidence has hindered the adoption of the UVV program in the UK and other European countries as this could lead to higher healthcare resource utilization. We thoroughly tested different exogenous boosting assumptions in our model. In the base case, our model used boosting assumptions based upon a 20-year real-world evidence study in UK adults conducted by Forbes et al. that showed the intensity of boosting to be lower (33%) than predicted by previous models [58]. Previously published models that included exogenous boosting generally predicted a transient increase in HZ in adults [16,18,38,59,60,61,62]. Our model also predicted a small rise in HZ incidence of between 402 and 409 cases per 100,000 person-years (a 5.3% to 7.1% increase compared to pre-UVV rates), which peaked 22 years after UVV implementation. However, there was an overall decrease in the incidence of HZ by 12.2% to 14.1% compared to the pre-UVV period after 50 years. This decrease in HZ incidence could be due to the diminishing cohort of individuals at risk for VZV reactivation due to the longer duration of the UVV program. Our model considered a dynamically changing population through inclusion of time-dependent mortality, fertility, and migration rates, using 50 years of historical data, making the model more realistic than a model with a static population. Our model showed a steady increase in HZ incidence even in the no-UVV scenario, reaching a peak of 408 cases per 100,000 person-years in 2062 (a 6.8% increase). This increase in HZ incidence in the absence of UVV could be primarily attributed to the ageing population. While this increase was comparable to the increase estimated for UVV strategies, the peak in the no-UVV scenario occurred at a later time period. 

The base-case model showed that all one- and two-dose UVV programs are cost-effective compared to no UVV, at a willingness-to-pay threshold of 20,000 GBP/QALY gained, either from the payer or the societal perspective. Further, the single-dose strategy with MMRV-MSD at 18 months is cost-saving from the societal perspective. If a UVV program is to be adopted, the cost-effectiveness frontier provides information on which strategy should be selected from a cost-effectiveness standpoint. Based on both the payer and societal perspective, only the single dose of MMRV-MSD at 18 months (Strategy A) was deemed cost-effective at the willingness-to-pay-threshold of GBP 20,000 per QALY gained. Two-dose strategies with catch-up, Strategy G (V-MSD 12 months, MMRV-MSD 40 months), and Strategy C (V-MSD at 12 and 18 months) lay on the frontier but are not cost-effective because their ICERs (for Strategy G vs Strategy A and for Strategy C vs Strategy G) exceeded the willingness-to-pay threshold of 20,000 GBP/QALY. The DSA suggests that the results are robust to changes in cost of medical care and vaccine coverage, take, waning, and vaccine cost. However, beyond the cost-effectiveness of vaccination strategies, policymakers should consider several factors when implementing a UVV program, including the ability to maintain high vaccination coverage (>80%) and the flexibility of the vaccination schedule to accommodate an additional vaccination visit, as well as the programmatic goals of the country. To prevent outbreaks and reduce transmission and breakthrough varicella, two-dose strategies are recommended over one-dose strategies [13]. 

The results of our model are consistent with those from a similar dynamic model adapted for Turkey, Italy, Mexico, Norway, and Switzerland [22,63,64,65,66]. These studies also showed significant reductions in burden of disease after UVV; however, only the Turkey, Norway, and Switzerland adaptations accounted for the impact of exogenous boosting. The effects of UVV on HZ incidence are strongly dependent on the hypothesized boosting intensity. A range of exogenous boosting frameworks have been used in previous modeling studies, from no boosting to full permanent immunity [67]. However, the effect of exogenous boosting is still uncertain due to differences in study populations and environments and limited pre- and post-UVV data. In addition, other investigators have reported that the projected transient increases in HZ are sensitive to several parameters whose values are uncertain [38]. Recent observational studies have shown that the impact of UVV on exogenous boosting may not be as significant as previously estimated [19,22]. To understand the impact of the exogenous boosting assumption on the cost-effectiveness of the UVV strategies, robust scenario analyses were conducted. Based on our model, the higher the prevalence of varicella, the more influential the exogenous boosting assumption will be on the number of HZ cases, and thus on the costs and QALYs. Consequently, the ICERs for the UVV strategies (from the payer perspective, GBP 6,809–15,079 per QALY gained versus no UVV) increased under scenarios EB1 and EB2 where the magnitude of boosting was higher and decreased considerably under scenario EB3 without boosting (0%). Even in strategy EB2 (with 100% boosting), the ICERs from the payer perspective remained below 20,000 GBP/QALY for 6 of the 10 vaccination strategies, and below 30,000 GBP/QALY for all 10 strategies. 

Brisson et al. predicted that UVV would reduce the burden of varicella in England and Wales [68]. However, these benefits would be offset by an increase in the incidence of HZ, and consequently, they reported that pediatric UVV might not be cost-effective. Another modeling study by Van Hoek et al. estimated that two-dose UVV would significantly reduce the burden of varicella with high vaccination coverage rates but with an increase in HZ incidence [18,41]. The authors evaluated the impact of a combined vaccination policy for varicella and HZ and concluded that the program would only be cost-effective over a long timeframe (80 to 100 years). A recent modeling study by Akpo et al. found two-dose UVV strategies to be cost-effective over short- and long-term time horizons [51]. The results from all these models were significantly influenced by the hypothesized impact of UVV on exogenous boosting and incidence of HZ, which were different in each model. Like our model, Melegaro et al. [61] also accounted for the impact of an ageing population and estimated a similar clinical impact on the burden of VZV [61]. They concluded that the concurrent introduction of routine HZ vaccination at 65 years of age with pediatric varicella vaccination is expected to mitigate the increase in HZ incidence and be a cost-effective policy in Italy [61]. We conducted a scenario analysis in which a HZ vaccination program (with Zostavax^®^ (Merck Sharp & Dohme LLC, Rahway, NJ, USA) was initiated in 2013 within the model. We observed a modest impact of HZ vaccination on HZ incidence and mortality with increased costs compared to no HZ vaccination. This could be due to the narrow age range (adults 70–80 years) that is recommended to receive HZ vaccination in the current program. Nevertheless, the ICERs for each UVV strategy observed in the base case without HZ vaccination were lower than the ICERs in the HZ vaccination scenarios. The HZ vaccination program could be more effective if it was offered to a broader age group of adults who are at a high risk of HZ infection, similar to recommendations in other countries such as the US [62].

### Limitations

This study has several limitations. Data specific to England and Wales for some parameters such as workdays lost and QALYs lost for HZ and PHN were not available; hence, we used published estimates from other countries in line with other studies. There is currently no pediatric vaccination visit at 18 months in England and Wales, which was the modeled timepoint for the one-dose MMRV strategies and two-dose short-interval strategies. The model did not include any costs associated with implementing (e.g., training staff) a new vaccination visit. Additionally, the list prices for MMRV formulations are not available for England and Wales; we used international reference pricing, which may differ from the actual list prices. Therefore, this is a conservative estimate of ICERs as the analysis utilizes list prices and not tender prices. There are limited data available comparing V-MSD/MMRV-MSD to V-GSK/MMRV-GSK; however, V-MSD and MMRV-MSD are generally considered immunologically equivalent to each other, as are V-GSK and MMRV-GSK. Finally, we used the temporary immunity model to estimate exogenous boosting and its duration. There is ongoing research on alternative modeling of the exogenous boosting mechanism, such as progressive immunity [67]. Different modeling approaches have led to different epidemiological results and would impact the cost-effectiveness of a UVV program.

## 5. Conclusions

We estimated the long-term effectiveness and cost-effectiveness of UVV in England and Wales while considering the impact on the burden of HZ using recent real-world evidence on exogenous boosting. We also used a dynamic population in our transmission model to better understand the role of an aging population on HZ incidence. 

Our model estimated that all UVV strategies, including one or two doses, short or medium intervals, and monovalent or quadrivalent formulations, substantially reduced the clinical burden of varicella in terms of incidence, outpatient visits, hospitalizations, and mortality compared to no UVV. A UVV program was projected to reduce the incidence of HZ after 50 years compared to a no-UVV scenario following an initial increase in the first 20 years of its introduction. Our model also showed a similar increase in the incidence of HZ in the absence of UVV because of the ageing population. All UVV strategies were cost-effective compared to no UVV over 50 years. A one-dose UVV strategy with MMRV-MSD administered at 18 months of age was the most dominant strategy from the payer perspective and was cost-saving from the societal perspective. Our model suggests that policymakers should consider UVV to reduce the clinical and economic burden of VZV in England and Wales. 

## Figures and Tables

**Figure 1 vaccines-10-01416-f001:**
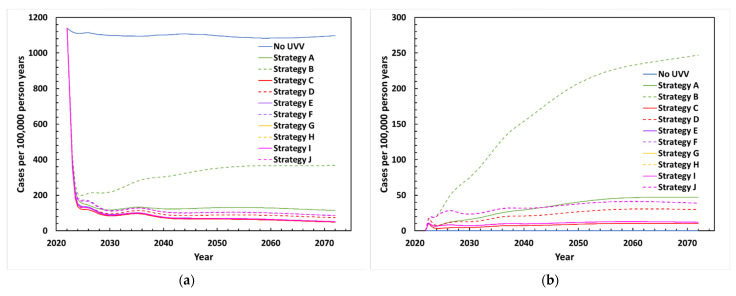
Projected varicella incidence between 2022 and 2072 with various UVV strategies: (**a**) Total varicella cases; (**b**) breakthrough varicella cases. **Strategy A**: MMRV-MSD (18 months); **Strategy B**: MMRV-GSK (18 months); **Strategy C**: V-MSD (12 months, 18 months, catchup); **Strategy D**: V-GSK (12 months, 18 months, catchup); **Strategy E**: V-MSD (12 months) + MMRV-MSD (18 months); **Strategy F**: V-GSK (12 months) + MMRV-GSK (18 months); **Strategy G**: V-MSD (12 months) + MMRV-MSD (40 months) + V-MSD (catchup); **Strategy H**: V-GSK (12 months) + MMRV-GSK (40 months) + V-GSK (catchup); **Strategy I**: V-MSD (12 months, 40 months, catchup); **Strategy J**: V-GSK (12 months, 40 months, catchup).

**Figure 2 vaccines-10-01416-f002:**
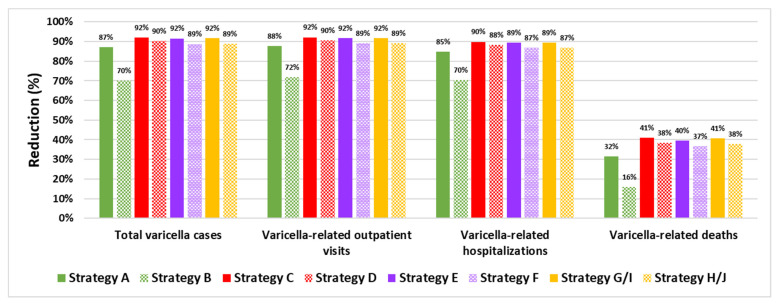
Percentage reduction in varicella health outcomes with various UVV strategies in comparison to a no-UVV strategy.

**Figure 3 vaccines-10-01416-f003:**
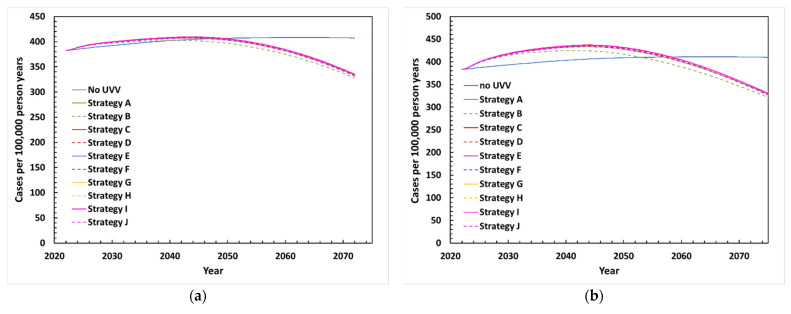
Projected incidence of HZ between 2022 and 2072 with various UVV strategies under (**a**) base-case scenario, (**b**) exogenous boosting scenario EB1 with waning period of HZ immunity of 24.4 years and age dependent effective contacts (75%, 71%, 57%, and 32% for <60, 60–69, 70–79, and ≥80 years old, respectively), (**c**) exogenous boosting scenario EB2 with waning period of HZ immunity of 81.3 years and 100% effective contacts, (**d**) exogenous boosting scenario EB3 with waning period of HZ immunity of 81.3 years and 0% effective contacts, (**e**) HZ vaccination scenario.

**Figure 4 vaccines-10-01416-f004:**
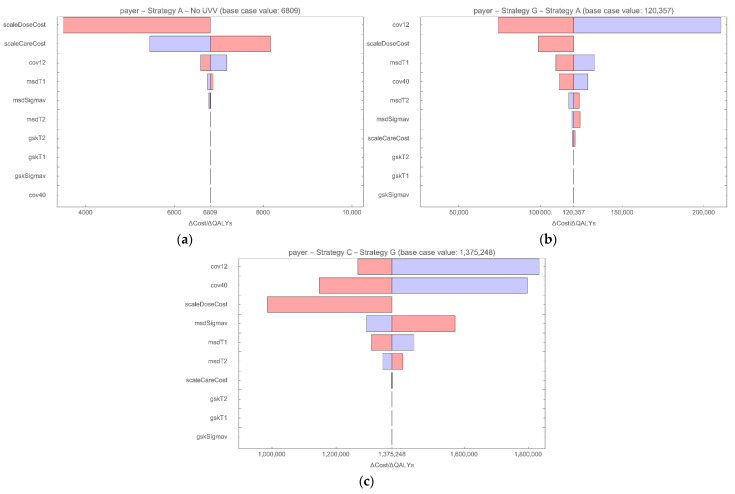
Tornado diagram DSA for strategies on the frontier from the payer perspective for (**a**) Strategy A vs. no UVV; (**b**) Strategy G vs. Strategy A; (**c**) Strategy C vs. Strategy G). ICERs based on lower bound (red) and upper bound (blue) parameter values. **cov12**: coverage rate for vaccination at month 12; **cov40**: coverage rate for vaccination at month 40; **gskSigmav**: GSK vaccine, average duration of protection; **gskT1**: GSK vaccine, 1st dose take; **gskT2**: GSK vaccine, 2nd dose take; **msdSigmav**: MSD vaccine, average duration of protection; **msdT1**: MSD vaccine, 1st dose take; **msdT2**: MSD vaccine, 2nd dose take; **scaleCareCost**: multiplier for disease-related costs; **scaleDoseCost**: multiplier for vaccine cost per dose.

**Table 1 vaccines-10-01416-t001:** Demographic and epidemiological parameters.

Parameter		Value	Source
*Demographic Rates, per year*			
$f_{j}\;\left(t\right)$	fertility rate	time-dependent	[27,28]
$\mu_{j}\;\left(t\right)$	force of mortality	time-dependent	[27,28,29]
$\Upsilon_{j}\;\left(t\right)$	force of migration	time-dependent	[30]
$d_{j}\;\left(t\right)$	maturation rate	time-dependent	computed
$\beta_{j_{1},j_{2}}$	contact mixing rates	mixing matrix	[31]
Λ (t)	birth rate	time-dependent	computed
*Natural Varicella Infection*			
1ωm	waning period of passive immunity	6 months	[23,32]
1ϵn	latent period of natural varicella	14 days	[23,33,34]
1γn	infectious period of natural varicella	7 days	[23,35]
1δn	waning period of HZ immunity	81.3 years	[19], Appendix A
*Breakthrough Varicella Infection*			
$\rho^{v}$	relative infectivity of breakthrough varicella	50%	[36]
1ϵvb	latent period of breakthrough varicella	14 days	[23,33]
1γvb	infectious period of breakthrough varicella	4.5 days	[23,35]
1δvb	waning period after breakthrough varicella of wild type HZ	81.3 years	[19], Appendix A
*Varicella Susceptibility*			
Δ	ramp	0.693099	calibrated
$rr_{1}$	susceptibility age < 5	0.153501	calibrated
$rr_{2}$	susceptibility 5 ≤ age < 10	0.0892316	calibrated
$rr_{3}$	susceptibility 10 ≤ age < 20	0.0713221	calibrated
$rr_{4}$	susceptibility 20 ≤ age	0.070527	calibrated
*HZ Reactivation*			
$\rho^{z}$	relative infectivity of HZ	7%	[23,37]
ω	wild HZ reactivation rate	0.228646	calibrated
ϕ	wild HZ reactivation rate	0.16355	calibrated
η	wild HZ reactivation rate	1.56705	calibrated
π	wild HZ reactivation rate	4.40466	calibrated
χ	reactivation rate factor on vaccine arms	1/6	[38]
1ηn	infectious period after natural varicella of wild-type HZ	28 days	[23,39]
1ηvb	duration of HZ outbreak following breakthrough varicella	28 days	[23]
1ηvv	infectious period following successful vaccination of wild-type HZ	28 days	[23]
1δvv	waning period of EB HZ immunity following successful vaccination	81.3 years	[19], Appendix A
*Contacts Leading to Exogenous Boosting*			
$\zeta^{n}$	after natural varicella	33.45%	[19], Appendix A
$\zeta^{vb}$	after breakthrough varicella	33.45%	[19], Appendix A
$\zeta^{vv}$	after varicella vaccination	33.45%	[19], Appendix A
*Disease-Related Death, by year of age group*			
$d_{j}^{v}$	varicella infection-related		See Appendix A
	0–5	0.00063%	
	5–15	0.00067%	
	15–45	0.01980%	
	45–65	0.14400%	
	≥65	1.54500%	
$d_{j}^{z}$	breakthrough varicella infection-related		[24]
	0–5	0.00020%	
	5–15	0.00020%	
	15–45	0.00018%	
	45–65	0.01500%	
	≥65	0.13800%	
$d_{j}^{z}$	HZ reactivation-related death rate		See Appendix A
	0–5	0.00000%	
	5–15	0.00100%	
	15–45	0.00240%	
	45–65	0.00360%	
	≥65	0.07900%	

**Table 2 vaccines-10-01416-t002:** Cost-effectiveness analysis for alternative UVV strategies from the payer and societal perspectives (2022–2072).

Strategy	Cost, GBP	QALYs Lost	Base Case Analysis (vs. No UVV)			Frontier Analysis		
			Incremental Costs, GBP	QALYs Gained	ICER, GBP	Incremental Costs, GBP	QALYs Gained	ICER, GBP/QALY
Payer perspective								
No UVV	1,717,399,462	375,267	-	-	-	-	-	On frontier
Strategy B	2,319,165,759	320,413	601,766,297	54,854	10,970	-	-	Strongly dominated
Strategy A	2,122,052,620	315,839	404,653,158	59,428	6809	404,653,158	59,428	6809
Strategy F	2,311,046,483	314,526	593,647,021	60,741	9773	-	-	Weakly dominated
Strategy J	2,637,071,252	314,276	919,671,790	60,991	15,079	-	-	Strongly dominated
Strategy H	2,311,390,487	314,276	593,991,025	60,991	9739	-	-	Weakly dominated
Strategy E	2,388,104,420	313,865	670,704,958	61,402	10,923	-	-	Weakly dominated
Strategy D	2,743,114,745	313,839	1,025,715,283	61,428	16,698	-	-	Strongly dominated
Strategy I	2,555,772,276	313,625	838,372,814	61,642	13,601	-	-	Strongly dominated
Strategy G	2,388,495,426	313,625	671,095,964	61,642	10,887	266,442,806	2214	120,357
Strategy C	2,641,172,957	313,441	923,773,495	61,826	14,942	252,677,531	184	1,375,338
Societal perspective								
No UVV	4,347,695,805	375,267	-	-	-	-	-	Strongly dominated
Strategy B	4,700,532,257	320,413	352,836,452	54,854	6432	-	-	Strongly dominated
Strategy A	4,261,892,370	315,839	−85,803,435	59,428	Dominates	-	-	On frontier
Strategy F	4,427,683,719	314,526	79,987,914	60,741	1317	-	-	Strongly dominated
Strategy J	4,745,295,541	314,276	397,599,736	60,991	6519	-	-	Strongly dominated
Strategy H	4,419,614,637	314,276	71,918,832	60,991	1179	-	-	Weakly dominated
Strategy E	4,463,708,494	313,865	116,012,689	61,402	1889	-	-	Strongly dominated
Strategy D	4,839,303,449	313,839	491,607,644	61,428	8003	-	-	Strongly dominated
Strategy I	4,622,322,562	313,625	274,626,757	61,642	4455	-	-	Strongly dominated
Strategy G	4,455,045,663	313,625	107,349,858	61,642	1742	193,153,293	2214	87,251
Strategy C	4,703,637,630	313,441	355,941,825	61,826	5757	248,591,967	184	1353,100

**Strategy A**: MMRV-MSD (18 months); **Strategy B**: MMRV-GSK (18 months); **Strategy C**: V-MSD (12 months, 18 months, catchup); **Strategy D**: V-GSK (12 months, 18 months, catchup); **Strategy E**: V-MSD (12 months) + MMRV-MSD (18 months); **Strategy F**: V-GSK (12 months) + MMRV-GSK (18 months); **Strategy G**: V-MSD (12 months) + MMRV-MSD (40 months) + V-MSD (catchup); **Strategy H**: V-GSK (12 months) + MMRV-GSK (40 months) + V-GSK (catchup); **Strategy I**: V-MSD (12 months, 40 months, catchup); **Strategy J**: V-GSK (12 months, 40 months, catchup).

**Table 3 vaccines-10-01416-t003:** Scenario analysis for the cost-effectiveness of alternative UVV strategies by perspective (2022–2072).

Strategy	Perspective-Specific Costs, GBP		QALYs Lost	Perspective-Specific Incremental Costs, GBP		Incremental QALYs	Perspective-Specific ICER, GBP/QALY	
	Payer	Societal		Payer	Societal		Payer	Societal
**Scenario EB1**								
No UVV	1,727,007,437	4,327,806,694	379,104	-	-	-	-	-
Strategy A	2,178,901,927	4,409,609,321	328,125	451,894,490	81,802,627	50,979	8864	1605
Strategy B	2,368,456,912	4,816,066,892	331,233	641,449,475	488,260,198	47,871	13,400	10,200
Strategy C	2,700,764,892	4,862,241,863	326,236	973,757,455	534,435,169	52,868	18,419	10,109
Strategy D	2,801,730,134	4,994,459,098	326,447	1,074,722,698	666,652,404	52,657	20,410	12,660
Strategy E	2,447,196,962	4,620,649,859	326,571	720,189,526	292,843,165	52,534	13,709	5574
Strategy F	2,368,826,858	4,579,699,466	326,975	641,819,421	251,892,772	52,130	12,312	4832
Strategy G	2,447,895,919	4,612,874,615	326,382	720,888,483	285,067,921	52,722	13,673	5407
Strategy H	2,369,470,088	4,572,521,358	326,774	642,462,651	244,714,664	52,330	12,277	4676
Strategy I	2,615,029,060	4,780,007,581	326,382	888,021,623	452,200,887	52,722	16,843	8577
Strategy J	2,694,716,188	4,897,767,515	326,774	967,708,751	569,960,821	52,330	18,492	10,892
**Scenario EB2**								
No UVV	1,683,646,322	4,222,564,986	368,848	-	-	-	-	-
Strategy A	2,185,741,631	4,475,908,293	327,148	502,095,309	253,343,307	41,700	12,041	6075
Strategy B	2,365,580,183	4,848,202,132	328,432	681,933,861	625,637,146	40,416	16,873	15,480
Strategy C	2,711,945,809	4,941,934,713	326,026	1,028,299,487	719,369,727	42,822	24,013	16,799
Strategy D	2,811,355,932	5,070,300,703	325,952	1,127,709,610	847,735,717	42,896	26,290	19,763
Strategy E	2,457,534,496	4,697,844,968	326,215	773,888,174	475,279,983	42,633	18,152	11,148
Strategy F	2,376,898,298	4,650,803,284	326,213	693,251,976	428,238,299	42,635	16,260	10,044
Strategy G	2,458,645,302	4,691,306,808	326,103	774,998,980	468,741,823	42,745	18,131	10,966
Strategy H	2,377,932,146	4,644,829,674	326,085	694,285,824	422,264,689	42,763	16,236	9875
Strategy I	2,625,681,749	4,858,343,181	326,103	942,035,427	635,778,195	42,745	22,039	14,874
Strategy J	2,702,905,856	4,969,803,371	326,085	1,019,259,534	747,238,386	42,763	23,835	17,474
**Scenario EB3**								
Strategy A	2,093,942,862	4,156,246,284	311,966	324,017,636	−352,175,909	74,820	4331	Dominates
Strategy B	2,302,213,635	4,635,830,833	318,800	532,288,408	127,408,640	67,986	7829	1874
Strategy C	2,609,122,561	4,585,839,215	308,808	839,197,335	77,417,022	77,978	10,762	993
Strategy D	2,712,436,180	4,724,782,352	309,483	942,510,954	216,360,158	77,303	12,192	2799
Strategy E	2,356,903,660	4,348,386,500	309,395	586,978,433	−160,035,693	77,391	7585	Dominates
Strategy F	2,281,901,408	4,317,913,358	310,467	511,976,182	−190,508,835	76,319	6708	Dominates
Strategy G	2,356,812,778	4,338,384,791	309,063	586,887,552	−170,037,403	77,723	7551	Dominates
Strategy H	2,281,773,045	4,308,487,644	310,127	511,847,819	−199,934,549	76,659	6677	Dominates
Strategy I	2,524,216,724	4,505,788,836	309,063	754,291,498	−2,633,358	77,723	9705	Dominates
Strategy J	2,607,853,149	4,634,567,492	310,127	837,927,923	126,145,299	76,659	10,931	1646
**Scenario HZV**								
No UVV	3,031,028,796	5,661,391,881	366,137	-	-	-	-	-
Strategy A	3,404,494,784	5,544,163,047	305,739	373,465,988	−117,228,834	60,398	6183	Dominates
Strategy B	3,606,506,838	5,987,831,352	310,502	575,478,042	326,439,471	55,636	10,344	5867
Strategy C	3,921,823,867	5,984,135,880	303,273	890,795,071	322,743,999	62,864	14,170	5134
Strategy D	4,024,412,076	6,120,435,662	303,698	993,383,280	459,043,781	62,439	15,910	7352
Strategy E	3,668,866,278	5,744,312,674	303,708	637,837,482	82,920,793	62,429	10,217	1328
Strategy F	3,592,815,553	5,709,278,538	304,403	561,786,757	47,886,657	61,734	9100	776
Strategy G	3,669,339,142	5,735,735,055	303,461	638,310,346	74,343,174	62,676	10,184	1186
Strategy H	3,593,228,993	5,701,283,370	304,148	562,200,197	39,891,489	61,989	9069	644
Strategy I	3,836,619,158	5,903,015,002	303,461	805,590,362	241,623,121	62,676	12,853	3855
Strategy J	3,918,920,871	6,026,973,826	304,148	887,892,075	365,581,945	61,989	14,323	5898
**Scenario Discount Rate (1.5% for health outcomes)**								
No UVV	1,717,399,462	4,347,695,805	568,030	-	-	-	-	-
Strategy A	2,122,052,620	4,261,892,370	476,226	404,653,158	−85,803,435	91,804	4408	Dominates
Strategy B	2,319,165,759	4,700,532,257	483,733	601,766,297	352,836,452	84,297	7139	4186
Strategy C	2,641,172,957	4,703,637,630	472,264	923,773,495	355,941,825	95,766	9646	3717
Strategy D	2,743,114,745	4,839,303,449	472,867	1,025,715,283	491,607,643	95,163	10,778	5166
Strategy E	2,388,104,420	4,463,708,494	472,872	670,704,958	116,012,689	95,158	7048	1219
Strategy F	2,311,046,483	4,427,683,719	473,822	593,647,021	79,987,914	94,208	6301	849
Strategy G	2,388,495,426	4,455,045,663	472,517	671,095,964	107,349,858	95,513	7026	1124
Strategy H	2,311,390,487	4,419,614,637	473,449	593,991,025	71,918,832	94,581	6280	760
Strategy I	2,555,772,276	4,622,322,562	472,517	838,372,814	274,626,757	95,513	8778	2875
Strategy J	2,637,071,252	4,745,295,541	473,449	919,671,790	397,599,735	94,581	9724	4204

**Scenario EB1**, 24.4 years waning period of HZ immunity; age-dependent % effective contact. **Scenario EB2**: 81.3 years waning period of HZ immunity; 100% effective contact. **Scenario EB3**: 81.3 years waning period of HZ immunity; 0% effective contacts. **Scenario HZV**: Vaccination against HZ. **Strategy A**: MMRV-MSD (18 months); **Strategy B**: MMRV-GSK (18 months); **Strategy C**: V-MSD (12 months, 18 months, catchup); **Strategy D**: V-GSK (12 months, 18 months, catchup); **Strategy E**: V-MSD (12 months) + MMRV-MSD (18 months); **Strategy F**: V-GSK (12 months) + MMRV-GSK (18 months); **Strategy G**: V-MSD (12 months) + MMRV-MSD (40 months) + V-MSD (catchup); **Strategy H**: V-GSK (12 months) + MMRV-GSK (40 months) + V-GSK (catchup); **Strategy I**: V-MSD (12 months, 40 months, catchup); **Strategy J**: V-GSK (12 months, 40 months, catchup).

## Data Availability

The datasets used and/or analyzed during the current study are available from the corresponding author on reasonable request.

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
