# Peer review of "Modeling the Impact of Exogenous Boosting and Universal Varicella Vaccination on the Clinical and Economic Burden of Varicella and Herpes Zoster in a Dynamic Population for England and Wales"

_vaccines, 2022, doi:10.3390/vaccines10091416_

Round 1

Reviewer 1 Report

Authors in this study estimated the long-term clinical and economical impact of UVV strategies in England and Wales and recommending exogenous boosting of the vaccine. The study is interesting and enough background is provided with all the required details. 

However, the figures are too blurry and hard to read. Vaccination strategies are not clearly defined as to why it was aligned with the MMR schedule? 

Overall, Materials and methods are clearly stated.

Author Response

Point 1: Figures are too blurry and hard to read.

Response 1: Thank you for your feedback regarding the figures.  TIF files have been created for each figure in the manuscript and supplement.  In addition, the size of the font within the figures were increased and darkened to improve clarity.

Point 2: Vaccination strategies are not clearly defined as to why it was aligned with the MMR schedule?

Response 2: Varicella and MMR vaccines are generally administered at same the visit provided their vaccination schedule overlaps (e.g., 1st dose 12 months and 2nd dose 3-5 years of age).  There is also option of using combination MMRV as quadrivalent vaccine. The following sentence was added (line 158-160):

“By aligning with MMR, the total number of pediatric office visits are not increased, and it also provides the opportunity to use combination MMRV vaccine in a single vaccine.”

Reviewer 2 Report

This is an important study with immense public health implications. Authors have used modelling techniques and used the recent RWE (on exogenous busting) to ascertain the cost-effectiveness of ten UVV strategies in England and Wales. I have a few observation regarding the method and reporting the results.

1. Please mention the source of health state utilities (HSU) used in the model that is reported in Suppl Table S10

2. Was the caretaker burden considered in the model? Since most hospital admission for varicella was in children under 10 and most HZ admission were for people over 60 years of age. Both these populations may require caretakers.

3. Isn't, NICE's recent recommendation suggests using a 3.5% discount for cost and 1.5% for the benefit for those interventions that have long-term effects, such as vaccines? At least the author can do sensitivity analysis with a 1.5% discount for effects. This is likely to improve the ICER of these strategies.

Source:

[PharmacoEconomics (2018) 36:745–758]

[Australian Health Review, 2020, 44, 337–339]

Author Response

Point 1. Please mention the source of health state utilities (HSU) used in the model that is reported in Suppl Table S10

Response 1: References for the health state utilities can be found in Table S10 in the column headers (references are 27-30). Full references are available at the end of supplement as well.

Point 2: Was the caretaker burden considered in the model? Since most hospital admission for varicella was in children under 10 and most HZ admission were for people over 60 years of age. Both these populations may require caretakers.

Response 2: Thank you for pointing out potential role of caretaker burden with both varicella and zoster.  Our model accounts for caregiver burden in terms of workdays lost. Inputs used for workdays lost by age group for varicella and HZ are shown in table S12.

Point 3. Isn't, NICE's recent recommendation suggests using a 3.5% discount for cost and 1.5% for the benefit for those interventions that have long-term effects, such as vaccines? At least the author can do sensitivity analysis with a 1.5% discount for effects. This is likely to improve the ICER of these strategies.

Response 3: You are correct that NICE recommends using differential discounting with 1.5% for interventions with long-term effects.  We have included this as a sensitivity analysis as suggested, adding the following text in the Methods (see lines 226-228) and Results (see Table 3 and text in lines 452-456).

Methods: “Finally, we conducted a sensitivity analysis using a 1.5% annual discount rate for long-term effects, consistent with NICE recommendations for vaccines [57].”

Results: “Finally, the results of the sensitivity analysis with a 1.5% discount rate for health outcomes are provided in Table 3.  Total costs for each strategy remained the same, but incremental QALYs increased.  Consequently, the ICERs for each strategy were lower than in the base case and ranged from $4,408 to $10,778 per QALY from the payer perspective and from cost saving to $5,166 per QALY from the societal perspective.”